# Drivers of inappropriate use of antimicrobials in South Asia: A systematic review of qualitative literature

Jennifer L. Murray[1,2]*, Daniel T. Leung[3], Olivia R. Hanson[3], Sharia M. Ahmed[3], Andrew T. Pavia[3], Ashraful I. Khan[4], Julia E. Szymczak[3], Valerie M. Vaughn[3], Payal K. Patel[5], Debashish Biswas[4], Melissa H. Watt[1]

1 Department of Population Health Sciences, University of Utah, Salt Lake City, Utah, United States of America, 2 School of Medicine, University of Utah, Salt Lake City, Utah, United States of America, 3 Department of Internal Medicine, University of Utah, Salt Lake City, Utah, United States of America, 4 International Centre for Diarrhoeal Disease Research, Bangladesh (icddr,b), Dhaka, Bangladesh, 5 Department of Internal Medicine, Intermountain Health, Murray, Utah, United States of America

* jennifer.murray@hsc.utah.edu

**Data Availability Statement:** All relevant data are within the paper.

**Funding:** This systematic review was accomplished with support from the National

## Abstract

Antimicrobial resistance is a global public health crisis. Effective antimicrobial stewardship requires an understanding of the factors and context that contribute to inappropriate use of antimicrobials. The goal of this qualitative systematic review was to synthesize themes across levels of the social ecological framework that drive inappropriate use of antimicrobials in South Asia. In September 2023, we conducted a systematic search using the electronic databases PubMed and Embase. Search terms, identified *a priori*, were related to research methods, topic, and geographic location. We identified 165 articles from the initial search and 8 upon reference review (n = 173); after removing duplicates and preprints (n = 12) and excluding those that did not meet eligibility criteria (n = 115), 46 articles were included in the review. We assessed methodological quality using the qualitative Critical Appraisal Skills Program checklist. The studies represented 6 countries in South Asia, and included data from patients, health care providers, community members, and policy makers. For each manuscript, we wrote a summary memo to extract the factors that impede antimicrobial stewardship. We coded memos using NVivo software; codes were organized by levels of the social ecological framework. Barriers were identified at multiple levels including the patient (self-treatment with antimicrobials; perceived value of antimicrobials), the provider (antimicrobials as a universal therapy; gaps in knowledge and skills; financial or reputational incentives), the clinical setting (lack of resources; poor regulation of the facility), the community (access to formal health care; informal drug vendors; social norms), and policy (absence of a regulatory framework; poor implementation of existing policies). This study is the first to succinctly identify a range of norms, behaviors, and policy contexts driving inappropriate use of antimicrobials in South Asia, emphasizing the importance of working across multiple sectors to design and implement approaches specific to the region.

Institutes of Health (R01 AI135114 to DTL; R21 HD109819 to DTL and AIK; K24 AI166087 to DTL) and the Agency for Healthcare Research and Quality (K08 HS086530 to VMV). The funders had no role in study design, data collection and analysis, decision to publish, or preparation of the manuscript.

**Competing interests:** The authors have declared that no competing interests exist.

## Introduction

The World Health Organization has identified antimicrobial resistance (AMR), the emergence and spread of pathogens resistant to antimicrobial agents, as one of the top ten global public health threats facing humanity [1]. AMR is driven largely by misuse and overuse of antimicrobial agents within both the medical and agricultural sectors, which has increased over the last two decades [2]. AMR threatens the efficacy of commonly used clinical antimicrobial agents, posing significant threats to human health. Infections associated with antimicrobial resistant bacteria, as opposed to a non-resistant form, confers two times the risk of a serious health outcome and three times the risk of mortality [3]. It is estimated that in 2019 AMR directly resulted in 1.2 million deaths and was a contributing factor in almost 5 million additional deaths worldwide [4]. If effective action to curb AMR development is not taken, it is estimated that by 2050, antimicrobial resistant diseases could result in 10 million deaths annually across the globe [5].

South Asia has seen a rapid increase in access to and use of antimicrobials, accompanied by a rise in AMR [6]. Health services in many South Asian countries are fragmented and rely on an uncoordinated mix of private and public services. These fragmented health systems provide a space in which inappropriate antimicrobial use can go unchecked and AMR flourishes [7]. The impact of AMR on human health in South Asia is profound. In India, nearly 60,000 newborns die each year as a direct result of AMR neonatal infections [8]. A 2021 study found that Bangladeshi children with bacteremia resistant to all first- and second-line treatments had an increased risk of death compared to those with susceptible bacteria [9].

Antimicrobial stewardship is a holistic approach to facilitating responsible use and protection of antimicrobial agents through the combined efforts of individuals, organizations, institutions, and policies [10]. The goal of stewardship is to reduce AMR and improve patient outcomes by ensuring that antimicrobials are used only when necessary, that appropriate antimicrobials are chosen considering the risk of AMR, and that antimicrobials are used for the minimal duration necessary to treat infection [11]. Antimicrobial stewardship programs typically focus on efforts in the health care system that promote the appropriate use of antimicrobials within a facility [12]. However, the World Health Organization acknowledges that particularly in low- and middle-income country settings, antimicrobial stewardship requires the participation and buy-in of both formal and informal health care providers, community members, and patients [13]. Reducing the inappropriate use of antimicrobials requires a change in human thought and behavior [14]; therefore, interventions to promote antimicrobial stewardship need to be informed by the behavioral, social, cultural, and structural factors that shape how people use antimicrobials [15]. Understanding the multi-level factors that contribute to inappropriate use of antimicrobials is a key step in designing strategies to combat AMR.

Qualitative methods (e.g. interviews, ethnography, focus groups) are well-suited to generate knowledge about the social determinants of antimicrobial overuse and the context in which AMR flourishes [16]. Qualitative data contain in-depth insight elicited from research subjects in their own words, offering novel understanding of the actionable drivers of antimicrobial misuse. Systematic reviews of qualitative research are valuable in synthesizing across studies to identify commonalities across studies that can efficiently inform the design and implementation of antimicrobial stewardship [17]. In this review paper, we aimed to synthesize themes across levels of the social ecological framework that drive inappropriate use of antimicrobials in South Asia. These findings can help identify areas for future research and intervention to prevent and mitigate AMR in South Asia.

## Materials and methods

### Overview and study criteria

We conducted a systematic review of the literature and used thematic synthesis [18] to integrate findings across qualitative studies. In conducting this review, we adhered to the Preferred Reporting Items for Systematic Reviews and Meta-analyses (PRISMA) guidelines [19], with attention to the unique requirements for reporting qualitative research as outlined in the Enhancing Transparency in Reporting the Synthesis of Qualitative Research (ENTREQ) statement [20].

Studies were eligible to be included in the systematic review if they met the following criteria: 1) English language research paper in a peer-reviewed journal, 2) used qualitative methods (including in-depth interviews, focus group discussions, or ethnographic observations), 3) reported qualitative themes related to factors driving inappropriate use of antimicrobials, or factors impeding antimicrobial stewardship, 4) reported data related to the human consumption of antimicrobials, and 5) reported data collected in South Asia (defined by the World Bank as Afghanistan, Bangladesh, Bhutan, India, Maldives, Nepal, Pakistan, Sri Lanka) [21]. Systematic reviews, opinion pieces, and editorials were excluded. There were no restrictions with respect to the date of publication.

### Search strategy and selection criteria

On September 18, 2023, we conducted a systematic search of the electronic databases PubMed and Embase. The search and selection processes were conducted by the first author (JLM) and repeated by the third author (OH) for consistency. We identified search terms related to research method, topic, and location (Table 1). Search results from both databases were downloaded to Zotero for review. After eliminating duplicates, we reviewed abstracts to eliminate papers that did not meet the inclusion criteria. The full texts of all remaining papers were then reviewed to confirm that they met inclusion criteria. The reference lists of included papers were reviewed to identify any additional papers that missed in our search. Discrepancies were resolved by consulting the last author (MHW).

**Table 1. Database search terms.**

| | Embase | | PubMed | |
|---|---|---|---|---|
| Criteria | Search Term | Number of Results | Search Term | Number of Results |
| Research Method | 'qualitative research'/exp OR 'qualitative research' | 133,158 | (qualitative research[MeSH Terms]) OR (qualitative[Text Word]) | 348,817 |
| Topic | 'antimicrobial stewardship'/exp OR 'antimicrobial stewardship' OR 'drug resistance'/exp OR 'drug resistance' | 532,195 | (((antimicrobial stewardship[MeSH Terms]) OR (drug resistance, microbial[MeSH Terms])) OR (antibiotic stewardship[Text Word])) OR (antibiotic resistance[Text Word]) | 215,357 |
| Location | 'Afghanistan' OR 'Bangladesh' OR 'Bhutan' OR 'India' OR 'Maldives' OR 'Nepal' OR 'Pakistan' OR 'Sri Lanka' | 1,651,737 | (((((((Afghanistan[Text Word]) OR (Bangladesh[Text Word])) OR (Bhutan[Text Word])) OR (India[Text Word])) OR (Maldives[Text Word])) OR (Nepal[Text Word])) OR (Pakistan[Text Word])) OR (Sri Lanka[Text Word]) | 264,901 |
| Combined Criteria | ('antimicrobial stewardship'/exp OR 'antimicrobial stewardship' OR 'drug resistance'/exp OR 'drug resistance') AND ('qualitative research'/exp OR 'qualitative research') AND ('afghanistan' OR 'bangladesh' OR 'bhutan' OR 'india' OR 'maldives' OR 'nepal' OR 'pakistan' OR 'sri lanka') | 90 | (((((antimicrobial stewardship[MeSH Terms]) OR (drug resistance, microbial[MeSH Terms])) OR (antibiotic stewardship[Text Word])) OR (antibiotic resistance[Text Word])) AND ((qualitative research[MeSH Terms]) OR (qualitative[Text Word]))) AND (((((((Afghanistan[Text Word]) OR (Bangladesh[Text Word])) OR (Bhutan[Text Word])) OR (India[Text Word])) OR (Maldives[Text Word])) OR (Nepal[Text Word])) OR (Pakistan[Text Word])) OR (Sri Lanka[Text Word])) | 75 |

## Data extraction and synthesis

Eligible mixed-methods and multi-site studies were adjusted to only include qualitative findings from World Bank defined South Asian countries in our analysis. We created a memo template to extract the themes reported in each study. The memos included information about the study, and a description of major themes that were reported across the levels of the social ecological framework [22]. The social ecological framework was chosen as an organizing framework to facilitate identification and organization of barriers across different levels of influence. Table 2 describes how we defined each level of the social ecological framework for the purpose of this study.

Each memo was reviewed by a second analyst for review and verification of the capture of findings from the original paper. Disagreements were resolved by consensus. The memos were uploaded into NVivo (version 12 Pro) qualitative data analysis software. Applied thematic analysis [23] was used to identify common themes across studies. We created a coding structure that included the levels of social ecological framework as the codes, and emerging themes under each level as the sub-codes. Two individuals dually coded all memos and met periodically to review and reconcile emerging codes. Discrepancies in coding were resolved by JM and MHW. After all memos were coded, the first author (JLM) and last author (MHW) met to review the coding structure, to merge and split codes as needed. Coding reports were analyzed, with reference to the source material as needed to synthesize and contextualize the findings.

## Critical appraisal

To assess the quality of the included studies, two individuals evaluated each paper using the Critical Appraisal Skills Program (CASP) for qualitative studies [24] and resolved any discrepancies by consensus upon consulting the first (JLM) and last (MHW) authors. Following our CASP review, all studies met quality criteria of research objective, appropriate qualitative methodology, research design, recruitment strategy, and data collection methods.

## Registration and protocol

The protocol for the systematic review was registered with PROSPERO (CRD42023456791). Available from: https://www.crd.york.ac.uk/prospero/display_record.php?ID=CRD42023456791.

**Table 2. Operationalization of the social ecological framework.**

| Level | Definition |
| --- | --- |
| Individual | Patients and their caregivers[1] |
| Interpersonal | Formal health care workers |
| Facility | Clinical settings |
| Community | Community practices and social norms, including informal drug vendors[2] |
| Policy | Governance and legislation |

[1] We defined a caregiver as an adult who provides assistance to someone who needs help taking care of themselves including children, the elderly, or individuals with an illness or disability.

[2] We defined informal drug vendors as individuals who practice allopathic medicine and sell antimicrobials without having completed formal training at an accredited institution with a defined curriculum.

## Results

### Included studies

The search and selection processes are summarized in Fig 1. The initial literature search yielded 165 results, and 8 additional publications were included upon reference review throughout the process for a total of 173 papers considered. After removing 11 duplicates and 1 preprint, 161 publications underwent abstract review, then 50 to full-text review. 46 publications were included in the final analysis [25–70], with publication dates ranging from 2010 to 2023. Table 3 includes a detailed description of the 46 publications that met criteria and were included in the final analysis.

The papers represent 6 of the 8 South Asian countries of interest. India had the highest study representation (n = 20), followed by Pakistan (n = 12), Bangladesh (n = 8), Nepal (n = 4), then Bhutan and Sri Lanka (n = 1 each). No studies were identified from the Maldives or Afghanistan. The studies used a wide variety of qualitative methodologies, the most common being individual interviews (n = 37) and focus group discussions (n = 13). Human subjects included patients, physicians (working in various specialties and sectors), nurses, pharmacists, community members (urban and rural, with various socioeconomic status), caregivers, government employees, national policy advisors, pharmaceutical industry staff, staff of nongovernmental healthcare organizations, international policy body representatives, students and faculty, drug vendors, and informal healthcare providers.

Data extraction and synthesis revealed themes at each level of the social ecological framework (Fig 2). Table 4 provides brief descriptions of each of the 12 themes that emerged.

### Individual level: Patients and their caregivers

Two major themes emerged at the individual level, representing the experiences and circumstances of patients and their caregivers: self-treatment with antimicrobials, and the perceived value of antimicrobials.

**Drivers of inappropriate use of antimicrobials.** Multiple studies reported that patients self-treat with antimicrobials due to multiple barriers, with socioeconomic status cited most frequently [27,28,30,45,48,53,70]. For example, patients living in rural villages may not have adequate funds to travel to formal health care facilities and obtain diagnostic testing and prescriptions [26,37,51]. Additionally, it was noted that patients and caregivers may be hesitant to seek medical care when they have "minor ailments," due to the increased practice of referring out to specialized physicians which costs more money, additional time, and travel [71–73]. Limited access to the formal health care system often leads patients to self-treat with antimicrobials when presenting with a wide variety of symptoms including diarrhea, stomach pain, cough, and fever by rationing antimicrobials [74–77]. A patient's decision to self-treat is informed by the perception that antimicrobials work based on prior therapeutic success with antimicrobials in themselves, family, or friends. Antimicrobials for self-treatment are obtained by receiving medication from a friend or family member, using leftover medication, or directly obtaining medication without a prescription from a pharmacy [65,74,75,78].

**Perceived value of antimicrobials.** A number of studies reported the perception held by patients that the receipt of an antimicrobial prescription indicates high quality medical care, as antimicrobials are perceived as offering an objective and rapid solution to their illnesses [75,77,79,80]. Overall, antimicrobials are perceived as powerful drugs that provide a quick solution to a range of ailments [81]. Studies attribute this attitude to a lack of knowledge not only surrounding antimicrobial resistance, but more generally around medicine, diagnostics, and treatments [28,79,82–85]. Additionally, caregivers report feeling a sense of emotional relief

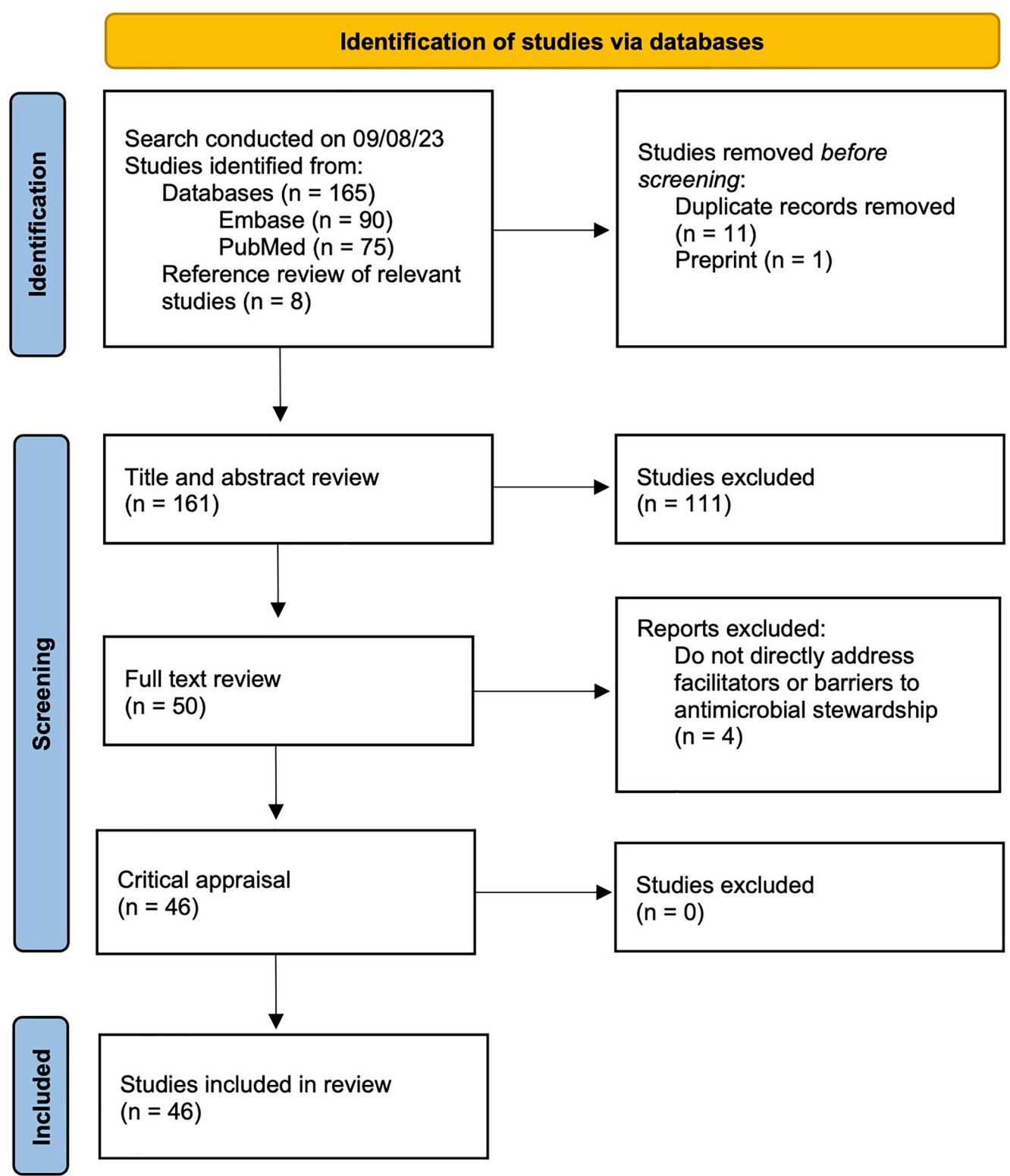

**Fig 1. PRISMA diagram showing the flow of study identification and selection.** Page MJ, McKenzie JE, Bossuyt PM, Boutron I, Hoffmann TC, Mulrow CD, et al. The PRISMA 2020 statement: an updated guideline for reporting systematic reviews. BMJ 2021;372:n71. 10.1136/bmj.n71.

**Table 3. Description of included publications.**

| First Author & Year | Title | Research Aim | Study Location | Qualitative Methods | Results: drivers of inappropriate use |
|---|---|---|---|---|---|
| Adhikari B 2021 [26] | Why do people purchase antibiotics over-the-counter? A qualitative study with patients, clinicians and dispensers in central, eastern and western Nepal | Explore characteristics and drivers of over the counter sales of antibiotics and implications for policy | Nepal | 28 individual interviews and 12 focus group discussions with dispensers at drug stores, patients, and clinicians | Patient demand, low access to formal care, inadequate and poor implementation of policy, social norms, informal community dispensers |
| Anwar M 2021 [35] | Exploring Nurses' Perception of Antibiotic Use and Resistance: A Qualitative Inquiry | To highlight nurses' perception of antibiotic use and resistance | Pakistan | 15 individual interviews with nurses | Patient demand, limited lab facilities, poor medical infrastructure, inadequate policy, lack of regulatory system |
| Atif M 2019 [59] | What drives inappropriate use of antibiotics? A mixed methods study from Bahawalpur, Pakistan | To investigate the knowledge, attitudes, and practices of the general public regarding the use of antibiotics in community pharmacy | Pakistan | 400 surveys and 16 individual interviews with patients and caregivers who have recently purchased antibiotics | Limited healthcare resources, accessibility, sharing prescriptions, non-adherence to treatment course, social norms |
| Atif M 2020 [32] | Community pharmacists as antibiotic stewards: A qualitative study exploring the current status of Antibiotic Stewardship Program in Bahawalpur, Pakistan | To assess community pharmacists' knowledge, perceptions, and current practices regarding the WHO Global Action Plan antibiotic stewardship program | Pakistan | 15 individual interviews with community pharmacists | Lack of education, stewardship program not utilized, lack of licensed pharmacists, lack of policy and regulations, lack of research, lack of public education |
| Atif M 2021 [27] | Antibiotic stewardship program in Pakistan: a multicenter qualitative study exploring medical doctors' knowledge, perception and practices | To investigate knowledge, perspectives, and practices of physicians regarding the antibiotic stewardship program | Pakistan | 17 individual interviews with doctors of tertiary care public sector hospitals | Patient demand, affordability, lack of education, limited healthcare resources, regulatory system not utilized |
| Barker A 2017 [54] | Social determinants of antibiotic misuse: a qualitative study of community members in Haryana, India | To assess the social determinants of antibiotic use among community members | India | 20 individual interviews with community members representative of a "typical" villager | Lack of education, affordability, limited healthcare resources, informal drug vendors |
| Barker A 2017 [54] | What drives inappropriate antibiotic dispensing? A mixed-methods study of pharmacy employee perspectives in Haryana, India | To better understand the factors that drive inappropriate antibiotic dispensing among pharmacy employees in India's village communities | India | 24 surveys and 20 individual interviews with community pharmacy employees | No regulation of retail pharmacists, lack of education, affordability, accessibility, limited healthcare resources, non-adherence to treatment course |
| Baubie K 2019 [61] | Evaluating antibiotic stewardship in a tertiary care hospital in Kerala, India: a qualitative interview study | To determine what barriers and facilitators to antibiotic stewardship exist within a 1300-bed tertiary care private hospital located in the state of Kerala, India | India | 31 semi-structured interviews and 4 focus groups with hospital staff ranging from physicians, nurses, pharmacists, and a clinical microbiologist | Accessibility, affordability, limited healthcare resources, lack of stewardship programs and regulatory policies |
| Biswas D 2020 [99] | An ethnographic exploration of diarrheal disease management in public hospitals in Bangladesh: From problems to solutions | To characterize challenges faced in diarrheal disease management in resource-limited hospitals and identify opportunities to improve care | Bangladesh | 138 individual interviews with clinicians, staff nurses and patients; rapid ethnographic observations | Reputational and financial incentive, lack of provider education, patient demand, understaffed hospitals, cultural barriers |
| Broom A 2020 [100] | Improvisation, therapeutic brokerage and antibiotic (mis)use in India: a qualitative interview study of Hyderabadi physicians and pharmacists | To explore social dynamics around antimicrobials and report on provider accounts of antimicrobial resistance | India | 3 interviews with 15 doctors and 15 pharmacists | Patient demand, self-medication, reputational and financial incentive, social norms, lack of regulation |
| Broom J 2021 [30] | Antimicrobial overuse in India: A symptom of broader societal issues including resource limitations and financial pressures | To identify experiences of antimicrobial prescribing and perceived barriers to optimal use | India | 30 individual interviews with 15 doctors, 15 pharmacists | Patient demand, non-adherence to treatment course, affordability, reputational and financial incentive, antimicrobials as a universal therapy, lack of regulation |

*(Continued)*

**Table 3.** (Continued)

| First Author & Year | Title | Research Aim | Study Location | Qualitative Methods | Results: drivers of inappropriate use |
|---|---|---|---|---|---|
| Chandy S 2013 [28] | Antibiotic use and resistance: perceptions and ethical challenges among doctors, pharmacists and the public in Vellore, South India | To inform stakeholder-targeted interventions to contain antibiotic use, gain support to address ethical issues, strengthen policy | India | 8 focus group discussions with urban and rural doctors, pharmacists, higher/lower socioeconomic public | Non-adherence to treatment course, lack of public education, affordability, patient demand, reputational and financial incentive, limited healthcare resources, social norms, lack of provider education |
| Charani E 2019 [41] | Investigating the cultural and contextual determinants of antimicrobial stewardship programmes across low, middle- and high-income countries—A qualitative study | To map the key contextual drivers of the development and implementation of antibiotic stewardship programs | India (multi-site) | 13 individual interviews from 7 hospitals | Limited capacity for providers to participate in stewardship programs, social norms, inadequate policy |
| Chowdhury M 2019 [53] | Rural community perceptions of antibiotic access and understanding of antimicrobial resistance: qualitative evidence from the Health and Demographic Surveillance System site in Matlab, Bangladesh | To explore factors and practices around access and use of antibiotics in rural communities with a socio-cultural perspective | Bangladesh | 6 focus group discussions and 16 individual interviews with residents | Self-medication, affordability, antimicrobials as a universal therapy, low patient trust of health systems, poor healthcare infrastructure, limited healthcare resources, convenient community pharmacies, sharing prescriptions |
| Darj E 2019 [51] | Pharmacists' perception of their challenges at work, focusing on antimicrobial resistance: a qualitative study from Bangladesh | To understand retail pharmacists' perceptions regarding antimicrobial resistance | Bangladesh | 24 individual interviews with retail pharmacists | Self-medication, non-adherence to treatment course, affordability, convenient community pharmacies, limited laboratory testing, lack of education, no regulation of retail pharmacists |
| Do N 2021 [33] | Community-based antibiotic access and use in six low-income and middle-income countries: a mixed-method approach | To compare community-based antibiotic access and use practices across communities in LMICs and identify contextually specific targets for interventions | Bangladesh (multi-site) | 16 individual interviews, 6 focus group discussions, 1100 surveys among drug suppliers and consumers | Accessibility, limited education, self-medication, sharing prescriptions, limited healthcare resources, convenient community pharmacies, lack of policy |
| Farooqui M 2023 [65] | Hospital Pharmacists' Viewpoint on Quality Use of Antibiotics and Resistance: A Qualitative Exploration from a Tertiary Care Hospital of Quetta City, Pakistan | To evaluate hospital pharmacists' understanding of antibiotic use and resistance at a public healthcare institute in Quetta city | Pakistan | 12 individual interviews with pharmacists | Accessibility, affordability, antimicrobials as a universal therapy, lack of patient education, lack of regulatory system and coordination |
| Hayat K 2019 [50] | Perspective of Pakistani Physicians towards Hospital Antimicrobial Stewardship Programs: A Multisite Exploratory Qualitative Study | To explore physician views of antimicrobial resistance and the benefit of hospital stewardship programs | Pakistan | 22 individual interviews with physicians from 7 tertiary care public hospitals | Non-adherence to treatment course, patient demand, lack of regulatory system, limited laboratory testing, no regulation of retail pharmacists, limited policy, limited enforcement of existing policy |
| Inchara M 2022 [66] | 'Perceptions' and 'practices' to antibiotic usage among diabetic patients receiving care from a rural tertiary care center: A mixed-methods study | To assess the various practices related to antibiotic use such as completion of prescribed schedule, re-use of prescriptions, over-the-counter purchase, use of leftover antibiotics in absence of medical advice, and others, and to explore the possible reasons for these practices in diabetic patients | India | 5 individual interviews with patients | Non-adherence to treatment course, lack of patient education, no regulation of retail pharmacists, limited healthcare resources |

*(Continued)*

**Table 3.** (Continued)

| First Author & Year | Title | Research Aim | Study Location | Qualitative Methods | Results: drivers of inappropriate use |
|---|---|---|---|---|---|
| Joseph H 2016 [62] | What Happens When "Germs Don't Get Killed and They Attack Again and Again": Perceptions of Antimicrobial Resistance in the Context of Diarrheal Disease Treatment Among Laypersons and Health-Care Providers in Karachi, Pakistan | To explore awareness of resistance, perceived causes, and potential solutions among a diverse sample of residents in a lower-middle-class community and a range of healthcare providers, including general practitioners, pharmacists and medical store owners, and unlicensed care providers | Pakistan | 85 semi-structured interviews with 40 laypersons and 45 healthcare providers in a lower-middle-class urban neighborhood in Karachi, Pakistan | Lack of education, non-adherence to treatment course, self-medication, no regulation of retail pharmacists |
| Kalam A 2021 [56] | Understanding the social drivers of antibiotic use during COVID-19 in Bangladesh: Implications for reduction of antimicrobial resistance | To identify the social drivers of antibiotic use among home-based patients who have tested positive or have COVID19-like symptoms | Bangladesh | 40 individual interviews with patients | Antimicrobials as a universal therapy, limited healthcare resources, limited laboratory testing, social norms, no regulation of retail pharmacists |
| Kalam A 2022 [86] | Antibiotics in the Community During the COVID-19 Pandemic: A Qualitative Study to Understand Users' Perspectives of Antibiotic Seeking and Consumption Behaviors in Bangladesh | To document how antibiotic are sought and used during COVID-19 and determine the reasons why patients may utilize these medicines sub-optimally | Bangladesh | 40 individual interviews with people diagnosed with or had symptoms suggestive of COVID-19 | Antimicrobials as a universal therapy, non-adherence to treatment course, doctor shopping, social norms, lack of education |
| Khan F 2021 [36] | Knowledge, Attitude, and Practice on Antibiotics and Its Resistance: A Two-Phase Mixed-Methods Online Study among Pakistani Community Pharmacists to Promote Rational Antibiotic Use | To investigate the knowledge, attitude, and practices of community pharmacists towards antibiotics, and to increase responsible use of antibiotics | Pakistan | 180 individual interviews with full and part-time community pharmacists | Self-medication, accessibility, patient demand, no regulatory system or coordination, no regulation of retail pharmacists, lack of policy |
| Khan F 2021 [36] | Exploring Undergraduate Pharmacy Students Perspectives Towards Antibiotics Use, Antibiotic Resistance, and Antibiotic Stewardship Programs Along With the Pharmacy Teachers' Perspectives: A Mixed-Methods Study From Pakistan | To investigate undergraduate students' level of knowledge related to antibiotic resistance and stewardship, and pharmacy faculty roles in role in ABR/AMS plans | Pakistan | 20 individual interviews with pharmacy teachers and 223 surveys of undergraduate pharmacy students from 12 universities | Lack of education and training for pharmacy graduates and students, no required stewardship curriculum, lack of stewardship program awareness |
| Khan F 2022 [79] | Evaluation of Consumers Perspective on the Consumption of Antibiotics, Antibiotic Resistance, and Recommendations to Improve the Rational use of Antibiotics: An Exploratory Qualitative Study From Post-Conflicted Region of Pakistan | To investigate the knowledge, attitude, and practices on antibiotic consumption, antibiotic resistance, and related suggestions of residents | Pakistan | 20 individual interviews with residents of conflicted zones | Lack of education, self-medication, antimicrobials as a universal therapy, limited healthcare resources, sharing prescriptions, no regulation of retail pharmacists, limited laboratory testing, lack of and limited regulation of policy |
| Khan M 2020 [43] | Is enhancing the professionalism of healthcare providers critical to tackling antimicrobial resistance in low- and middle-income countries? | To investigate whether weaknesses in health care providers' professionalism result in boundaries between "qualified" and "unqualified" providers being blurred, and how these weaknesses impact inappropriate provision of antibiotics in LMICs | Pakistan (multi-site) | 85 individual interviews (39 in Pakistan) with providers, government health agencies, national AMR technical policy advisers, pharmaceutical industry staff, healthcare NGOs, and local representatives of international policy bodies | No regulation of retail pharmacists, lack of education, no regulatory system or coordination, financial and reputational incentive, social norms |

(Continued)

**Table 3.** (Continued)

| First Author & Year | Title | Research Aim | Study Location | Qualitative Methods | Results: drivers of inappropriate use |
|---|---|---|---|---|---|
| Khare S 2022 [55] | Understanding Internal and External Drivers Influencing the Prescribing Behaviour of Informal Healthcare Providers with Emphasis on Antibiotics in Rural India: A Qualitative Study | To analyze the internal and external drivers that influence informal health care providers' prescribing behaviors for common illnesses in children under five, especially in rural areas | India | 7 focus group discussions with 48 informal health care providers | Limited healthcare resources, patient demand, lack of education, reputational incentives, lack of regulation or policy |
| Kotwani A 2010 [37] | Factors influencing primary care physicians to prescribe antibiotics in Delhi India | To explore the factors that influence primary care physicians to prescribe antibiotics and to investigate possible interventions | India | 3 focus group discussions with 36 primary care physicians in the public and private sectors | Patient demand, non-adherence to treatment course, doctor shopping, limited laboratory testing, limited healthcare resources, reputational and financial incentives, no regulation of retail pharmacists, little enforcement of existing regulations |
| Kotwani A 2012 [42] | Irrational use of antibiotics and role of the pharmacist: an insight from a qualitative study in New Delhi, India | To understand the dispensing practices and behavior of community pharmacists to develop policy interventions that would improve the use of antibiotics at the community level | India | 3 focus group discussions with 40 retail pharmacists, public sector pharmacists, and the office bearers of pharmacists' associations | Self-medication, financial incentives, accessibility, lack of laboratory testing, no regulation of retail pharmacies |
| Kotwani A 2016 [46] | Knowledge and perceptions on antibiotic use and resistance among high school students and teachers in New Delhi, India: A qualitative study | To explore the perceptions and knowledge of schoolteachers and students about antibiotic use, resistance, and suggestions for practical interventions | India | 5 focus group discussions with high school students (years 9–11) and 5 with teachers from private and public schools | Self-medication, lack of education, antimicrobials as a universal therapy, non-adherence to treatment course, limited healthcare resources, financial incentives, lack of policy enforcement |
| Kotwani A 2017 [52] | Prescriber and Dispenser Perceptions About Antibiotic Use in Acute Uncomplicated Childhood Diarrhea and Upper Respiratory Tract Infection in New Delhi: Qualitative Study | To explore the prescribing practices, knowledge, and attitudes of primary care doctors and community pharmacists, regarding antibiotic use in acute upper respiratory tract infections (URTI) and diarrhea in children to better understand causes of misuse and identify opportunities and suggestions to change behaviors | India | 2 focus group discussions (8–12 participants) with primary care government doctors, private general practitioners, pediatricians, and community pharmacists. 22 individual semi-structured interviews with various providers | Self-medication, patient demand, non-adherence to treatment course, antimicrobials as a universal therapy, reputational incentives, limited healthcare resources, limited laboratory testing, no regulation of retail pharmacists |
| Kotwani A 2021 [48] | Knowledge and behavior of consumers towards the non-prescription purchase of antibiotics: An insight from a qualitative study from New Delhi, India | To investigate the knowledge, practice, and behavior of consumers towards antibiotics, resistance, and purchasing behaviors, and to gain insight to inform evidence-based policy interventions | India | 72 individual interviews in 11 districts | Self-medication, non-adherence to treatment course, affordability, antimicrobials as a universal therapy, limited healthcare resources, lack of education, reputational incentives, no regulation of retail pharmacists |
| Kotwani A 2021 [72] | Over-the-Counter Sale of Antibiotics in India: A Qualitative Study of Providers' Perspectives across Two States | To gain insight into the OTC sale of antibiotics at retail pharmacies and explain its underlying drivers | India | 36 individual interviews with 22 pharmacists and 14 informal dispensers from 36 retail pharmacies | Self-medication, accessibility, affordability, antimicrobials as a universal therapy, reputational and financial incentives, limited healthcare resources, no regulation of retail pharmacists, lack of education, lack of policy, little enforcement of existing regulations |

(Continued)

**Table 3.** (Continued)

| First Author & Year | Title | Research Aim | Study Location | Qualitative Methods | Results: drivers of inappropriate use |
|---|---|---|---|---|---|
| Kotwani A 2023 [67] | Strengthening antimicrobial stewardship activities in secondary and primary public healthcare facilities in India: Insights from a qualitative study with stakeholders | To examine the existing federal and state policies in place that could strengthen antimicrobial stewardship activities in district and sub-district hospitals in India | India | Individual interviews with 15 national/state policy makers and stakeholders who implement policy, such as physicians and administration | Limited enforcement of existing policies and regulation, lack of education, limited knowledge of hospital stewardship programs, limited laboratory testing |
| Lucas P 2019 [49] | Pathways to antibiotics in Bangladesh: A qualitative study investigating how and when households access medicine including antibiotics for humans or animals when they are ill | To explore how households in Bangladesh were accessing antimicrobials for themselves and their domestic animals | Bangladesh | Individual interviews with 48 urban and rural households | Affordability, accessibility, limited healthcare resources, no regulation of retail pharmacists, social norms, sharing of prescriptions |
| Maki G 2020 [38] | Feasibility Study of the World Health Organization Health Care Facility-Based Antimicrobial Stewardship Toolkit for Low- and Middle-Income Countries | To obtain local input on the WHO health facility antimicrobial stewardship toolkit content and implementation of the steward program | Nepal and Bhutan (multi-site) | Individual interviews with 12 policy makers, 21 hospital administrators, 20 physicians, 21 nurses, 11 pharmacists, and 13 laboratory techs | Limited healthcare resources and personnel, lack of provider education, limited implementation of stewardship programs, limited financial support and personnel for stewardship programs, lack of regulation enforcement, lack of policy |
| Mitchell J 2023 [68] | Exploring the potential for children to act on antimicrobial resistance in Nepal: Valuable insights from secondary analysis of qualitative data | To consider the specific roles children and young people play in AMR-driving behaviors | Nepal | Secondary analysis of transcript data from focus group discussions and individual interviews with 23 adults engaged in an AMR-focused film project | Lack of education, little age-inclusive community engagement and education measures, non-adherence to treatment course |
| Nahar P 2020 [57] | What contributes to inappropriate antibiotic dispensing among qualified and unqualified healthcare providers in Bangladesh? A qualitative study | To explore knowledge surrounding the use and functions of antibiotics, awareness of antimicrobial resistance, and perceived patient or customer demand and adherence among providers | Bangladesh | 46 individual interviews with urban and rural community health care providers | Antimicrobials as a universal therapy, limited education, no regulation of retail pharmacists |
| Nair M 2019 [58] | "Without antibiotics, I cannot treat": A qualitative study of antibiotic use in Paschim Bardhaman district of West Bengal, India | To explore the drivers of antibiotic use among formal/informal healthcare providers and patients accessing care at primary health centers | India | 28 individual interviews with allopathic doctors, informal health providers, nurses, pharmacy shopkeepers, and patients | Self-medication, non-adherence to treatment course, accessibility, reputational and financial incentives, limited laboratory testing, no regulation of retail pharmacists |
| Nair M 2023 [69] | Perceptions of effective policy interventions and strategies to address antibiotic misuse within primary healthcare in India: A qualitative study | To assess perceptions of interventions and gaps in policy and practice with respect to outpatient antibiotic misuse in India | India | 23 individual interviews with individuals in academia, non-government organizations, policy, advocacy, pharmacy, and medicine | Limited research, lack of structural changes to address socio-ecological drivers, lack of policy, limited healthcare infrastructure and resources |
| Pearson M 2019 [44] | Knowing antimicrobial resistance in practice: a multi-country qualitative study with human and animal healthcare professionals | To investigate AMR awareness among human/animal healthcare professionals and the contextual issues influencing the relationship between awareness and practices of antimicrobial prescribing and dispensing | India (multi-site) | Individual interviews and rapid ethnographic observation with healthcare professionals | Antimicrobials as a universal therapy, lack of healthcare resources, poor community hygiene and sanitation |
| Rolfe R 2021 [31] | Barriers to implementing antimicrobial stewardship programs in three low and middle-income country tertiary care settings: findings from a multi-site qualitative study | To determine perceived barriers to the development and implementation of antibiotic stewardship programs in tertiary care centers in three LMICs | Sri Lanka (multi-site) | Individual interviews with 22 physicians at tertiary care hospitals | Patient demand, limited laboratory testing, limited healthcare resources, lack of education, accessibility, social norms, lack of policy |

*(Continued)*

**Table 3.** (Continued)

| First Author & Year | Title | Research Aim | Study Location | Qualitative Methods | Results: drivers of inappropriate use |
|---|---|---|---|---|---|
| Sahoo K 2010 [63] | Antibiotic use, resistance development and environmental factors: a qualitative study among healthcare professionals in Orissa, India | To explore physician, veterinarian, and drug dispenser opinions about antibiotic use and antibiotic resistance development in relation to environmental factors | India | Individual interviews with 24 physicians, veterinarians, and drug dispensers in Orissa | Climate variability and pollution, social norms, accessibility, affordability, limited healthcare resources, lack of policy |
| Saleem Z 2019 [64] | Antimicrobial prescribing and determinants of antimicrobial resistance: a qualitative study among physicians in Pakistan | To assess physician perception about antibiotic use and resistance, and factors influencing their prescribing of antibiotics and potential interventions to improve their future antibiotic prescribing | Pakistan | Individual interviews with 15 physicians registered with the Pakistan Medical and Dental Association (PMDC) | Lack of education, limited implementation of hygienic measures, limited healthcare resources, reputational and financial incentives |
| Shrestha A 2023 [10] | The Resistance Patterns in E. coli Isolates among Apparently Healthy Adults and Local Drivers of Antimicrobial Resistance: A Mixed-Methods Study in a Suburban Area of Nepal | To describe the antimicrobial resistance pattern in E. coli isolated from the fecal samples of apparently healthy individuals in Dhulikhel municipality and to explored the local drivers of AMR | Nepal | Focus group discussions and individual interviews with pharmacy workers, food vendors, health coordinators, and community members in Dhulikhel | Lack of education, no regulation of retail pharmacists, non-adherence to treatment course, limited healthcare resources, affordability, lack of policy |
| Singh S 2021 [40] | Investigating infection management and antimicrobial stewardship in surgery: a qualitative study from India and South Africa | To investigate the drivers for infection management and antimicrobial stewardship across high-infection-risk surgical pathways | India (multi-site) | Case-studies, ethnographic observations, and individual interviews with 44 healthcare professionals and 6 patients | Patient demand, limited physician capacity to practice stewardship guidelines, no regulatory system or coordination |

receiving an antimicrobial prescription because they believe it will successfully treat their children, elders, and other vulnerable groups [81,86]. The high value placed on antimicrobials and belief in their power leads patients to expect and even demand antimicrobials when seeking medical care [87–89]. Patients will "doctor shop" or seek care only from providers who are known to readily provide antimicrobials or pharmacies that are liberal with their distribution [65,90–92].

## Interpersonal level: Formal health care workers

Three major themes emerged at the interpersonal level, representing formal health care workers: antimicrobials as a universal therapy, gaps in knowledge and skills, and financial or reputational incentives.

**Antimicrobials as a universal therapy.** In multiple studies, participants spoke about how antimicrobials provide a cheap and accessible treatment plan for a wide variety of medical conditions, particularly in the absence of diagnostic and treatment options [71,82,90]. Clinician participants described that it is common practice for providers to prescribe antimicrobials when they are unsure of a patient's medical diagnosis, waiting for laboratory testing results, or even as a preventative measure to reduce the occurrence of secondary infections [71,89,93]. Dispensing antimicrobials based on prior therapeutic success was described as an appropriate treatment for patients presenting with similar symptoms [82,89]. Therefore, in many countries broad-spectrum antimicrobials was perceived as universal therapy for any general illness in conjunction with other common medications such as ibuprofen or acetaminophen [71,72,77,82,89]. Some providers directly handed medications to patients without writing a prescription or providing the medication name [77].

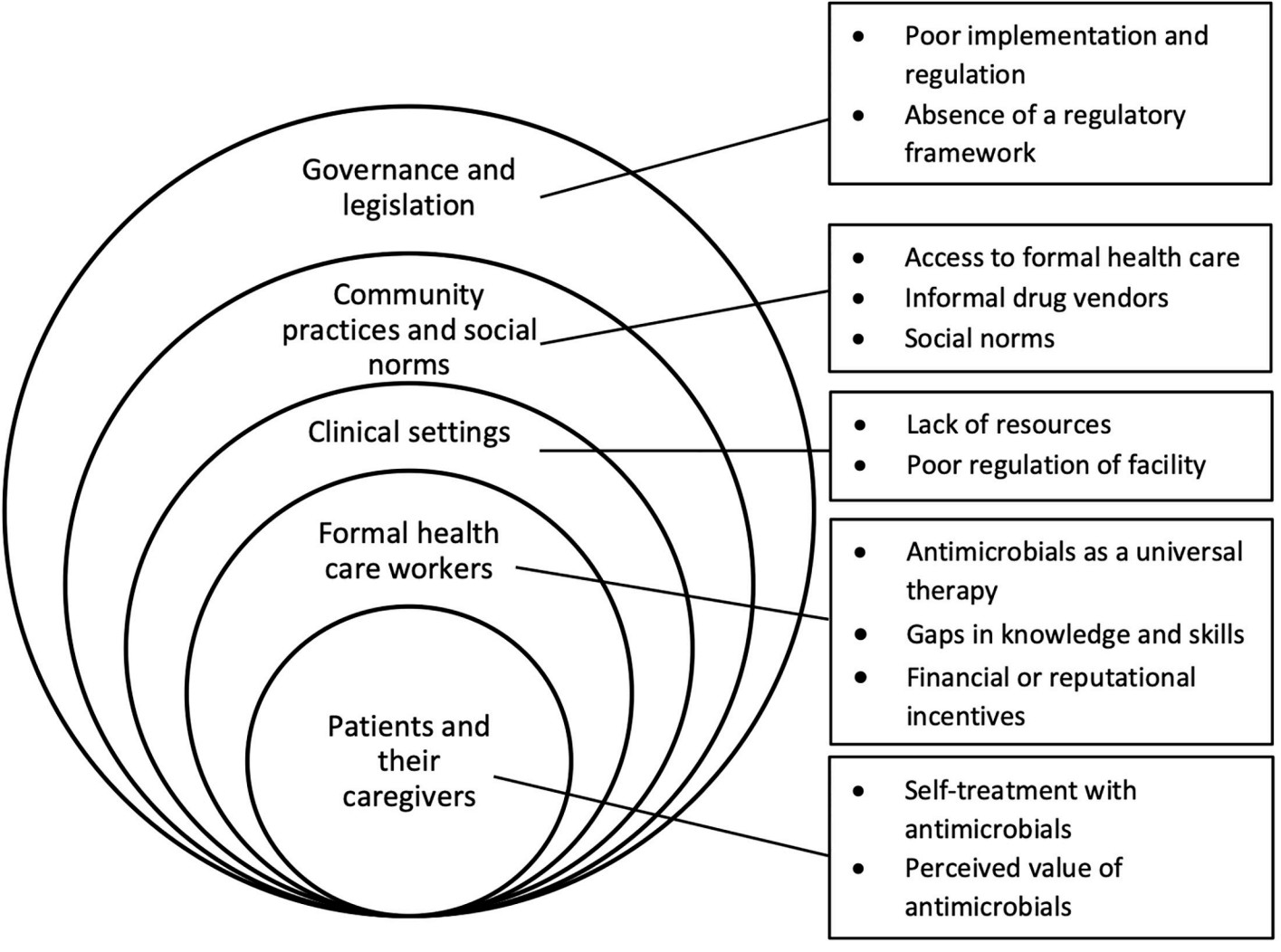

**Fig 2. Summary of themes across the social ecological framework.**

**Gaps in knowledge and skills.** Many health care workers reported lacking knowledge surrounding antimicrobial stewardship and appropriate prescribing practices. Multiple studies with clinicians and students noted a lack of awareness of existing antimicrobial stewardship programs in their facilities, and an absence of training curricula on appropriate use of antimicrobials [94,95]. Among clinicians with some understanding of antimicrobial resistance and/or stewardship programs, many held misconceptions or denied the severity of the problem [89,96]. Others believed that specific medical specialties (e.g., surgery) or individuals in leadership (e.g., chief physicians) should be responsible for taking action and managing antimicrobial distribution in their own teams and departments [97,98].

**Financial or reputational incentives.** Clinicians described the pressures they faced to dispense antibiotics to their patients. They noted that patients associated the dispensing of medications with a high level of care, leading them to dispense antimicrobials in hopes of increasing business and patient satisfaction [28,77,90]. Business success is dependent on positive community reviews and reputation, leading clinicians to prioritize patient demands over clinical guidelines, especially with wealthy or influential patients [28,99]. It was reported that pharmaceutical

**Table 4. Emerging themes and short descriptions.**

| Level | Theme | Description |
|---|---|---|
| *Individual: Patients and their caregivers* | **Self-treatment with antimicrobials** | Patients forego formal health care and obtain antimicrobials through various means (e.g., purchasing antimicrobials without a prescription, using leftover antimicrobials, taking family and friends' prescriptions). |
| | **Perceived value of antimicrobials** | Patients believe that usage of antimicrobials will cure them and alleviate their symptoms quickly. Antimicrobials are viewed as a universally effective drug for all symptoms/illnesses and are actively requested in clinical consultations. |
| *Interpersonal: Formal health care workers* | **Antimicrobials as a universal therapy** | Physicians inappropriately recommend antimicrobial therapy to patients as a first step in treatment, often in the absence of diagnostics. |
| | **Gaps in knowledge and skills** | Many providers lack training or knowledge of antimicrobial stewardship. Those who do have knowledge of these issues may not translate these ideas into clinical practice. |
| | **Financial or reputational incentives** | Providers are likely to prescribe antimicrobials to receive financial incentive from pharmaceutical companies or retain patients by maintaining a reputation of being willing to prescribe antimicrobials. |
| *Facility: Clinical settings* | **Lack of resources** | Lack of resources within facilities (e.g. space, time, providers, knowledge of programs, infrastructure, funding, etc.) that undermines quality care and leads to overuse of antimicrobials. |
| | **Poor regulation of facility** | Facility procedures are not properly implemented or regulated. (e.g., drug reporting procedures, management of medication, pluralistic health care services, organization of the healthcare facility, etc.) |
| *Community: Community practices and social norms* | **Access to formal healthcare** | Structural barriers prevent patients accessing formal health care in their communities. In contrast, informal providers are perceived as relatively easy, timely, and cost-effective, and readily provide antimicrobials. |
| | **Informal drug vendors** | Pharmacies, medical shops, and informal health care providers in the wider community provide antimicrobials to patients without requiring lab testing, office visits, formal prescriptions, etc. |
| | **Social norms** | The use and sharing of antimicrobials are widely accepted as a normal and common practice. |
| *Policy: Governance and legislation* | **Absence of a regulatory framework** | Lack of policy and implementation frameworks addressing proper antimicrobial use, stewardship, and prescribing practices. |
| | **Poor implementation and regulation** | Existing policies are poorly implemented or not regulated by the government. |

companies also play a role by providing financial incentives to providers for prescribing high volumes of certain antimicrobials and other medications [28,69,87,90,99–101].

## Facility level: Clinical settings

Two major themes emerged at the facility level, representing formal health care facilities: lack of resources, and poor regulation of the facility.

**Lack of resources.** Multiple studies reported a widespread shortage of medical infrastructure, equipment, and personnel across diverse settings, resulting in poor access to laboratory testing and diagnostics. In urban settings, there is a shortage of hospital personnel paired with a high volume of patients, leading to the prescription of common antimicrobial regimens without in-depth assessment of patients or laboratory testing [82,88,95,99,102,103]. Additionally, physicians are often unavailable or there are long wait times, prompting individuals to obtain antimicrobials on their own rather than access these formal facilities [28,66,77,84]. Multiple studies also reported that in rural areas, treatment and testing facilities are lacking altogether, requiring patients to travel long distances to reach health facilities. As a response to these resource shortages in the formal health care system, informal medical practices and drug stores are common, where individuals may directly purchase antimicrobials without a prescription [84].

**Poor regulation of the facility.** Studies showed conflicting accounts of antimicrobial availability and regulation within facilities. In India, Kotwani et al. report that public sector

healthcare facilities will under- or over-prescribe antimicrobials based on their current stock, leading to inconsistent prescription patterns that encourage patients to share or obtain antimicrobials from community and family members [76]. When antimicrobials are formally prescribed, facilities often lack or underuse drug reporting systems and do not maintain clinical documentation. Several studies noted the challenges of developing and implementing antimicrobial stewardship program, depending largely on the prioritization of leadership [81,97]. The hierarchical structure of medical systems can be a barrier in implementing antimicrobial stewardship practices if senior physicians or leadership does not prioritize it [97].

## Community level: Community practices and social norms

Three major themes emerged at the community level: access to formal healthcare, informal drug vendors, and social norms.

**Access to formal healthcare.**   Studies described a multitude of systemic barriers to accessing the formal health care system, including rural areas with limited infrastructure, long wait times, poor quality of care, an inability to pay for services, or the complexity of navigating health systems [73,74,87]. Instead of accessing formal care, many patients reported that they instead relied on informal drug vendors who ran small drug stores that provided consultations and dispensed medications [65,72,74,75].

**Informal drug vendors.**   These informal drug vendors report being commercially driven to sell medications and reach sales targets, often resulting in an over prescription of antimicrobials, inaccurate dosing, and the distribution of "half" antimicrobials which may be mixed with other materials (e.g., caffeine, routine pain medications) [28,74,85]. In a study of informal healthcare providers in rural India, Khare et al. explained how informal drug vendors are an essential resource for rural and medically underserved communities, where antimicrobials are often handed out in response to a verbal description of symptoms or patient demand based on prior treatment success [91]. The authors noted that without informal drug vendors, these patients would likely forego healthcare entirely.

**Social norms.**   Social norms surrounding the use of antimicrobials also emerged as a significant factor promoting inappropriate use. The sharing of antimicrobials between family and friends is a socially accepted and rooted practice in many South Asian settings [74]. Additionally, individuals will often trade medical advice with their social networks and encourage others to obtain specific antimicrobials that treated their own symptoms in the past [86]. In Bangladesh, Lucas et al. explained that women will typically ask their husbands for diagnostic or treatment advice rather than visiting a formal physician [73]. Community norms related to antimicrobial use drive dispensing patterns. As mentioned at the interpersonal level, over-prescription of antimicrobials is common to retain patients and provide the perception of high-quality care that is associated with antimicrobials [28,102].

## Policy level: Governance and legislation

Two major themes emerged at the policy level: absence of a regulatory framework to monitor and control antimicrobials, and poor implementation of existing policies of antimicrobial stewardship.

**Absence of a regulatory framework.**   Studies across various countries noted that national, state, and local governments, and their policy infrastructures, were ultimately responsible for antimicrobial stewardship programs (ASPs) in clinical settings. In a study of ASP development and implementation in India, Charani et al. (2019) conducted interviews with clinical providers in India and noted a lack of national infrastructure to legislate and control access to antimicrobials; strong local leadership and championing was necessary to make up for this

shortcoming and create successful ASPs [98]. Similar concerns about regulation and surveillance were identified in Pakistan among physicians [50] and pharmacists [65]. The study by Hayat et al (2019) indicated several barriers in ASP implementation in hospitals, which could be overcome if the government were to provide necessary support, including legislation and funding [50].

**Poor implementation and regulation.**   Other studies noted that even in countries with existing government regulation and legislation, it is difficult to navigate, understand, and consistently enforce these policies in clinical settings [67,69,70]. In a study of policymakers and clinicians in Bhutan and Nepal, Maki et al. 2020 described policies related to prescription-only sales of antimicrobials, but a lack of enforcement in both the clinical and community settings [95].

## Discussion

We report the results of a qualitative systematic review of studies conducted in South Asian countries to examine the factors that drive inappropriate use of antimicrobials. Through the synthesis of findings reported in 46 qualitative studies, we identified multiple factors across five levels of the social ecological framework: the individual patient, the formal provider, the clinical setting, the community, and policy. Drivers of inappropriate use of antimicrobials were evident at all levels, highlighting the importance of working across multiple levels and sectors to address drivers of antimicrobial misuse and build commitment for stewardship in South Asia. These findings align with other systematic reviews and analyses of both qualitative and quantitative research on antimicrobial resistance and stewardship efforts around the world [104–106], emphasizing the need for coordinated global action in addition to region-specific solutions.

The heterogeneity of South Asian healthcare systems presents significant barriers to antimicrobial stewardship. The studies in our analysis described regional differences in facility sizes and accessibility, administration involvement, government influence, licensure and formal education requirements of healthcare workers, pharmacy policies, and drug regulation programs. Further research is needed to assess these factors in countries that were under-represented in the literature, such as Afghanistan and the Maldives, so interventions can be specifically tailored by region. Most studies depicted fragmented systems in which there is little communication amongst formal providers within individual hospitals and clinics, health systems, and their greater communities. Improving this communication is crucial for the success of any intervention. Compared to studies in Sub-Saharan Africa and Latin America, the utilization of informal drug dispensers and unregulated community pharmacies is much more prevalent in South Asia [104,107]. Therefore, it is essential for antimicrobial stewardship efforts in South Asia to target both formal and informal healthcare workers.

Both formal prescribers and informal drug dispensers face immense social and financial pressures from patients and pharmaceutical companies to liberally supply antimicrobials despite knowing about AMR and the resulting health consequences. Our data suggest that it is normative in many hospitals and clinics to order antimicrobials as a universal therapy to cover a variety of potential illnesses, appease patients, and generate pharmaceutical revenue. Shifting norms that are so embedded in the healthcare industry will require a multifaceted, longitudinal approach that encourages provider behavioral change. Potential solutions may include investing in the workforce to remove profit incentive of dispensing drugs, implementation of a regulatory framework to control antimicrobial prescribing in both public and private facilities, and required educational curriculum specific to antimicrobial stewardship in early stages of medical training to facilitate a sense of ownership and responsibility as providers. Community pharmacies and informal drug dispensers should also be formally regulated to control the use

of antimicrobials, though alternative opportunities for business revenue must be identified to encourage meaningful and sustainable change.

Patient expectations and demands were universally identified as a significant driver of inappropriate antimicrobial use. This is primarily driven by a larger systemic issue of healthcare inaccessibility, pushing individuals to demand antimicrobials during limited provider visits or seeking them in their communities and social networks instead. To effectively shift this cultural norm, increasing accessibility to care must be prioritized, especially in low-income and geographically isolated communities. Efforts might include investing in public transportation that extends to rural areas and villages, investing in education at all levels, and recruiting medical workers from underrepresented regions who are likely to return to those communities to practice. Existing facilities should expand to formally integrate laboratory testing and diagnostic equipment and should prioritize quality improvement to better serve patients. Additionally, public health campaigns and community health workers can better educate the general public on infection prevention and the negative impacts of antimicrobial overuse as has successfully been done in the Indian state of Kerala [108].

Our data suggested a dearth of policies addressing antimicrobial stewardship, and poor enforcement of existing policies. Interventions might include adopting a national antimicrobial monitoring system, requiring consults with pharmacists who have antimicrobial stewardship-specific expertise, providing financial incentives for infection prevention and reduced antimicrobial prescribing, or developing and requiring a national, standardized educational training for all antimicrobial dispensers [109–112]. Policy reform and legislation alone are not sufficient to facilitate widespread antimicrobial stewardship nor combat resistance; it is also necessary to change individual behaviors and the embedded cultural norms that encourage them. There are multiple public health programs that facilitate social and behavioral change at the individual, family, and community level [113]. For example, water, sanitation and hygiene (WASH) activities are administered by local, regional, national, and international groups and are even embedded in national education curricula in China, the Democratic Republic of the Congo, Nicaragua, and Sudan [114]. These efforts are successful due to a massive global coordination and multi-sector participation in WASH activities. Similar efforts must be made for antimicrobial stewardship to address the global health threat of antimicrobial resistance and its devastating health effects.

It's important to note some limitations of the study. First, this review focused on overuse and misuse of antimicrobials in human populations, and did not include other significant drivers of AMR, such as antimicrobial use in livestock [115], environmental changes [116], and water and sanitation systems [117,118]. A review of AMR in South Asia in a One Health framework [119] would be a valuable addition to the literature. Second, given the unequal representation of South Asian countries in the existing literature, these findings may not be generalizable to all of South Asia. The limited literature could be in part due to bias in research funding and publication. We only included studies published in English, potentially excluding studies that are otherwise eligible; however, we did not identify any manuscripts in the two databases that were published in a language other than English. Finally, although there was an established and detailed search methodology, it is possible that published studies that fit criteria were missed. However, we are confident that thematic saturation was reached as clear and consistent themes emerged across the included manuscripts.

## Conclusions

Antimicrobial resistance is a major threat to individual and population health in South Asia. As common antimicrobials become less efficacious due to antimicrobial-resistant organisms,

there is a risk of significant increases in morbidity and mortality in the region. In synthesizing the qualitative literature in South Africa, we identified a range of norms, behaviors, and policy contexts that contribute to antimicrobial resistance in South Asia. The findings point to a need for a multi-pronged approach that works across sectors to improve the surveillance and reporting of antimicrobial use and implement stewardship interventions specific to the unique regions.

## Supporting information

**S1 Checklist. PRISMA checklist.**
(DOCX)

## Acknowledgments

The following individuals helped to support the review process: Nehal Bakshi, Anya Tiwari, and Maya Stephens.

## Author Contributions

**Conceptualization:** Jennifer L. Murray.

**Data curation:** Jennifer L. Murray.

**Formal analysis:** Jennifer L. Murray, Melissa H. Watt.

**Funding acquisition:** Daniel T. Leung.

**Methodology:** Jennifer L. Murray, Melissa H. Watt.

**Supervision:** Melissa H. Watt.

**Visualization:** Jennifer L. Murray.

**Writing – original draft:** Jennifer L. Murray, Olivia R. Hanson, Melissa H. Watt.

**Writing – review & editing:** Jennifer L. Murray, Daniel T. Leung, Sharia M. Ahmed, Andrew T. Pavia, Ashraful I. Khan, Julia E. Szymczak, Valerie M. Vaughn, Payal K. Patel, Debashish Biswas, Melissa H. Watt.

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
