## [Decision Letter · Decision Letter 0]

7 Nov 2023

PGPH-D-23-01910

Drivers of inappropriate use of antimicrobials in South Asia: A systematic review of qualitative literature

Dear Dr. Jennifer L Murray,

Thank you for submitting your manuscript to PLOS Global Public Health. After careful consideration, we feel that it has merit but does not fully meet PLOS Global Public Health’s publication criteria as it currently stands. Therefore, we invite you to submit a revised version of the manuscript that addresses the points raised during the review process.

One of the reviewers suggested that the review would have benefited from using a scoping review approach compared to a systematic review. Provide a rationale why this was the best approach or consider suggesting the lack of grey literature as a limitation.

We look forward to receiving your revised manuscript.

Kind regards,

Ferdinand Mukumbang, PhD

Academic Editor

Journal Requirements:

1. Please provide separate figure files in .tif or .eps format only and remove any figures embedded in your manuscript file. Please also ensure all files are under our size limit of 10MB.

2. We have noticed that you have uploaded Supporting Information files, but you have not included a list of legends. Please add a full list of legends for your Supporting Information files after the references list.

Additional Editor Comments (if provided):

Reviewers' comments:

Reviewer's Responses to Questions

**Comments to the Author**

1. Does this manuscript meet PLOS Global Public Health’s publication criteria? Is the manuscript technically sound, and do the data support the conclusions? The manuscript must describe methodologically and ethically rigorous research with conclusions that are appropriately drawn based on the data presented.

Reviewer #1: Yes

Reviewer #2: Yes

2. Has the statistical analysis been performed appropriately and rigorously?

Reviewer #1: I don't know

Reviewer #2: N/A

3. Have the authors made all data underlying the findings in their manuscript fully available (please refer to the Data Availability Statement at the start of the manuscript PDF file)?

Reviewer #1: Yes

Reviewer #2: Yes

4. Is the manuscript presented in an intelligible fashion and written in standard English?

Reviewer #1: Yes

Reviewer #2: Yes

5. Review Comments to the Author

Reviewer #1: I want to thank the authors for writing a review on this hidden pandemic (AMR) in South Asia.

Abstracts:

Conclusion is missing. Also, consider using a structured format for abstract writing in this scenario.

Introduction

Well written

Methods

-Instead of overview, I suggest authors say overview and study criteria

-You mention only articles published in English were included, and is that not a bias?

-What was the review period? Who did the data search, when, how were full articles accessed if not open access, and why were only two databases searched?

-How were mixed-method studies handled?

-Critical appraisal should come after data extraction and how was consensus done for discrepancies?

-The quality check stage is missing from the PRISMA table

-Table 3: I recommend a column added stating main findings of the studies included in the final review

Results

a. Included studies

-Can authors explain what happened at each stage of the PRISMA table and why?

Reviewer #2: Thank you for the opportunity to review this manuscript. I want to commend the authors for taking on this important topic. However, I have some comments and questions:

1. In my opinion, the topic can benefit from a scoping or narrative review instead of systematic review.

2. Inclusion of grey literature such as government reports can be beneficial as they may also contain some valuable information.

3. The rationale regarding what this research adds to the knowledge is not clear. Other than the setting, the findings may not be new and already known. Other studies in LMICs have found similar results.

4. For search terms- why 'drug resistance' was used instead of 'antimicrobial resistance', which is used more commonly in this field? more synonyms of 'antimicrobial stewardship' to broaden search terms may increase the original pool of articles.

5. Inclusion of CINAHL, MEDLINE etc. libraries may offer more comprehensive literature search, which may have more qualitative articles than PubMed and Embase.

6. How findings were extracted when articles included multiple sites (e.g. studies across LMICs, studies including South Asia and another region).

7. Finally, the results section can be broken down in subtopics to help the reader in understanding sub themes within broader themes.

6. PLOS authors have the option to publish the peer review history of their article (what does this mean?). If published, this will include your full peer review and any attached files.

**Do you want your identity to be public for this peer review?** For information about this choice, including consent withdrawal, please see our Privacy Policy.

Reviewer #1: No

Reviewer #2: No

---

## [Decision Letter · Decision Letter 1]

21 Feb 2024

Drivers of inappropriate use of antimicrobials in South Asia: A systematic review of qualitative literature

PGPH-D-23-01910R1

Dear Dr Murray,

We are pleased to inform you that your manuscript 'Drivers of inappropriate use of antimicrobials in South Asia: A systematic review of qualitative literature' has been provisionally accepted for publication in PLOS Global Public Health.

Best regards,

Ferdinand Mukumbang, PhD

Academic Editor

Reviewer Comments (if any, and for reference):

Reviewer's Responses to Questions

**Comments to the Author**

1. If the authors have adequately addressed your comments raised in a previous round of review and you feel that this manuscript is now acceptable for publication, you may indicate that here to bypass the “Comments to the Author” section, enter your conflict of interest statement in the “Confidential to Editor” section, and submit your "Accept" recommendation.

Reviewer #2: All comments have been addressed

2. Does this manuscript meet PLOS Global Public Health’s publication criteria? Is the manuscript technically sound, and do the data support the conclusions? The manuscript must describe methodologically and ethically rigorous research with conclusions that are appropriately drawn based on the data presented.

Reviewer #2: Yes

3. Has the statistical analysis been performed appropriately and rigorously?

Reviewer #2: N/A

4. Have the authors made all data underlying the findings in their manuscript fully available (please refer to the Data Availability Statement at the start of the manuscript PDF file)?

Reviewer #2: Yes

5. Is the manuscript presented in an intelligible fashion and written in standard English?

Reviewer #2: Yes

6. Review Comments to the Author

Reviewer #2: (No Response)

7. PLOS authors have the option to publish the peer review history of their article (what does this mean?). If published, this will include your full peer review and any attached files.

**Do you want your identity to be public for this peer review?** For information about this choice, including consent withdrawal, please see our Privacy Policy.

Reviewer #2: No
